# Remote Diagnosis of Architectural Heritage Based on 5W1H Model-Based Metadata in Virtual Reality

**Jongwook Lee [1], Junki Kim [2], Jaehong Ahn [1] and Woontack Woo [1,***

[1]   Graduate School of Culture Technology, Korea Advanced Institute of Science and Technology, 291 Daehak-ro, Yuseong-gu, Daejeon 34141, Korea

[2]   Culture Technology Research Institute, Korea Advanced Institute of Science and Technology, 291 Daehak-ro, Yuseong-gu, Daejeon 34141, Korea

*   Correspondence: wwoo@kaist.ac.kr

**Abstract:** We propose a framework based on the 5W1H model-based metadata for remote diagnosis in virtual reality (VR). For this purpose, we suggest unique metadata composed of Point of Interest (POI)-extended anchor (xAnchor)-content for a context-aware service in virtual and augmented reality. We define the attributes of the metadata based on the 5W1H context for information retrieval according to the context in a remote diagnosis. Second, we propose the ontology-based linker metadata that express the relations between AR scenes and that retrieve external information. Moreover, we suggest heritage building information metadata for information retrieval according to context. For evaluation, we created a geo-tagged content tool and a remote diagnosis VR application. We conducted focus-group interviews and heuristic evaluations for remote diagnosis in VR to verify the methodology of this study. As a result, we found that experts were most satisfied with the functions that provide the contextualized information. This study contributes to the geospatial metadata for a context-aware service in VR/AR as well as the remote diagnosis framework to overcome the time-consuming problem of the existing remote diagnosis.

**Keywords:** metadata; virtual reality; remote diagnosis

## 1. Introduction

We propose a remote diagnosis framework based on the 5W1H (what, when, where, who, why, and how) model-based metadata in virtual reality (VR). There have been numerous studies on metadata for heritage management [1,2] and standardization of virtual and augmented reality metadata [3]. However, these metadata are not suitable for a VR-based remote diagnosis which provides on-site and related information according to the context in VR. For remote diagnosis in VR, we suggest Point of Interest (POI)- extended anchor (xAnchor)-content metadata structure and metadata based on the 5W1H model-based metadata in VR. The model sorts complex contexts into six categories. Therefore, it has an advantage in generating various situations and in reflecting user context for the information retrieval system according to the context [4]. Also, we propose the linker metadata for reflecting domain knowledge in VR application. Moreover, we introduce heritage building information metadata in case of risk management information. Finally, we suggest a remote diagnosis framework which transfers on-site and related information, such as components and past management information, to remote experts. Our methodologies enable a remote expert to effectively verify on-site and relevant information according to context and support to decide a risk response in VR.

Existing diagnosis of architectural heritage was conducted by an expert visiting the heritage site daily or regularly. This method not only is costly and time-consuming but also cannot manage thousands of architectural heritages due to the small number of experts. A remote diagnosis, which shares

on-site photos, drawings, checklists, and reports, has been conducted to deal with the above concerns. However, the solution also has a limitation in referring and sharing related information because they are scattered through various formats and sources. In order to overcome the above problems, many researchers have proposed to integrate the management information of architectural heritage and to share various experts' opinions using Historic Building Information Modeling (HBIM) [5,6]. However, the HBIM is not suitable to conduct a remote diagnosis because it cannot provide an immersive environment to diagnose the architectural building.

Several researchers have proposed to identify the conditions of architectural heritage using virtual reality [7,8]. Virtual reality has an advantage in simulating a real environment to give the impression of being immersed in that environment [9]. In the virtual environment, it is advantageous to check information about objects by using natural user interfaces, such as rotation and zooming in/out. For this reason, VR is a useful technology to check related information. Additionally, remote diagnosis in VR has the merit of collecting, confirming, and sharing information at the remote site. However, it is time-consuming for maintenance technicians to find the information required for risk management because the information does not have metadata appropriate for remote diagnosis in VR.

We suggest the 5W1H model-based metadata structure and metadata for remote diagnosis in VR. First, we propose metadata composed of POI-xAnchor-content for context-aware services in VR/AR. We define attributes of the metadata based on the 5W1H context model for information retrieval according to context. Second, we propose the linker metadata, which is a domain ontology-based metadata that expresses the relations between on-site information, historical data, sensing data and information, and the cause–effect relationships among components. In addition, we propose heritage building information metadata for information retrieval according to context. We introduce a remote diagnosis framework based on the 5W1H model-based metadata in VR.

The proposed method allows the site manager to input the description and the tag on the geo-tagged contents so that the maintenance technician receives retrieved and filtered contents. For this purpose, some of the metadata is utilized as a tag to transmit the status information of cultural heritage to maintenance technicians. They can identify the on-site information uploaded by the site manager and the general person with the remote diagnosis framework in VR. They can choose a POI that is colored green, yellow, and red, depending on the state. Experts can obtain information related to architectural heritage from a geo-tagged database. They can upload the diagnosis results as a checklist. In the future, the repairperson will implement management measures by referring to the uploaded diagnosis results.

To verify the methodology of this study, we conducted focus-group interviews and heuristic evaluations on the remote diagnosis in VR. We could get the whole consensus from the focus-group interviews that, even if a maintenance technician does not visit the site, remote diagnosis can produce results similar to those in the field. In heuristic evaluations, experts were most satisfied with the contextual information provided by the system.

In this study, we proposed the 5W1H model-based metadata, which enables information retrieval and content filtering according to the context in VR. The linker metadata reflect domain knowledge and support to provide related information and to visualize relationships among VR/AR scenes. We suggest a remote VR application that provides on-site and relevant information so that an expert can diagnose architectural heritage and decide risk responses at the remote site. This application enables the reduction of information overload of maintenance technicians by providing information according to the context.

## 2. Related Works

### 2.1. VR/AR Metadata Standard

Most VR/AR metadata contain geospatial information because VR/AR content is represented in the virtual world or real world. Each of VR standards focus on describing 3-D models, such as

mesh, material, and lights realized as Virtual Reality Modeling Language (VRML) and ARAF [3]. Augmented Reality Markup Language (ARML) 2.0 is a set of extensible standards and frameworks that can provide AR usage cases for the storage and sharing interoperability of AR contents [10]. Moreover, the 5W1H-based metadata provide a context-aware-mediated AR service that secures extensibility for further contextual information. The metadata structure ensures the reusability of the AR contents and interoperability of AR applications [11]. Context-Aware Mobile Augmented Reality (CAMAR) also provides contents by integrating contexts based on the 5W1H model and user profiles [12].

However, existing AR metadata has three problems in implementing context-aware AR services. First, there is no element to define relations among AR contents and it is difficult to provide related contents in context-aware service. Second, existing AR metadata does not contain information about the target user and the accessibility of contents, which makes it difficult to design personalized services. Third, sensor information required for outdoor AR service is not defined in detail. This study proposes the metadata structure to overcome the above problems and proves it through a prototype.

The metadata structure consists of a POI, content, and the xAnchor element. It is designed to solve problems that can be used only in a specific domain and to enhance scalability. We also propose the POI linker and the content linker metadata that represent semantic relations between content elements and xAnchor elements and explain the description of relations in detail. The proposed metadata structure can contribute to providing context-aware AR services by defining elements, property, and semantic relations necessary for context-aware AR services outdoor.

## 2.2. Virtual Reality for Risk Management

The cultural heritage domain is a sector that has always exploited the potential of VR technology. VR applications in the cultural heritage domain are meant for preservation, documentation, research, education, reconstruction, and exploration. For example, VR application for reconstruction means viewing and interacting with reconstructed architectural models and historical information in a virtual environment. In the architectural heritage field, VR technology is a valid cognitive tool and is a fundamental medium through which a user (a scholar, a student, or simply somebody who shows an interest for the subject) can interact with 3-D models and agents in a virtual environment [9]. VR technology has been primarily applied to share and visualize information intuitively and to provide the latest 3-D model information for heritage buildings [7,13].

VR technology in the field of risk management of architectural heritage also has some advantages; first, it allows people to interact, to intuitively share critical information, and to schedule repair operations. Second, it enables many professionals to conduct collaborative work in a virtual environment [14]. Few VR applications have dealt with the use of VR and interactions for the risk management of architectural heritage [7,15,16]. For example, Fassi (2016) developed an application that allows heritage managers to identify and share the risk information in a virtual environment [7].

A VR application linked to an HBIM system is useful for risk information management. HBIM data have been utilized in VR since the 2010s to address issues such as heterogeneous data sharing and communication between site managers and maintenance technicians [17,18]. Because HBIM includes various information such as historical document, monitoring data, and conservation state in conjunction with the database which can be integrated into a VR environment, it allows users to integrate, manage, and retrieve the necessary information. For example, Barazzetti (2015) introduced Autodesk A360 application based on HBIM, a commercial mobile VR application that supports building information searches and commenting on virtual buildings. In this application, professional operators could use A360 to exploit 2-D and 3-D drawings, object properties, and reports [16].

In this research, HBIM data include context information through the proposed metadata. It enables searching and sharing contextualized information in VR. Existing commercial VR applications cannot support the provision of contextualized information according to the context. For example, A360 supports the visualization of virtual models and the provision of information on the virtual models in VR but cannot support information retrieval according to the context. However, if the VR

application supports the retrieval of information according to the context in a virtual environment, it can shorten time and effort on risk management.

## 2.3. Augmented Reality for Risk Management

AR allows the user to see the real world, with virtual objects superimposed upon or composited with the real world [19]. AR technologies have been migrating from marker-based methods to markerless methods and recently to mobile context-aware methods, which can bring AR into the mobile and field context. Augmented reality technology has become widely used as data communication through a network in various domains according to the development of the mobile device and network technology.

In the cultural heritage domain, heritage information is provided in the form of texts and 3-D models through mobile devices to understand the archeological site and architectural heritage. Prieto (2017) classified three types of AR application for heritage. They are a reconstruction of dilapidated buildings, a recreation of archaeological missing or damaged parts, and simulation of social or natural environments on archaeological sites [20]. There were important projects for AR application: TIMEFRAME (1997), ARCHEOGUIDE (2000), Vilars (2001), and LIFEPLUS (2002).

AR can integrate, retrieve, and provide contextual information of analytical results using noninvasive tools directly in the field [21]. For disaster management, AR can be a solution where computer-generated information is superimposed over the real world, providing sufficient information and guidance [22]. Therefore, heritage managers can record the building condition and can intuitively compare the data. AR systems were proposed to support on-site risk management. For example, Chionna (2015) suggested an AR application to integrate the basic information regarding architectural heritage and correlate investigational data, such as thermal information using markerless AR technology [21]. Wallgrun (2018) superimposed changed environment features in an AR environment [23]. In terms of risk management, such as remote diagnosis, Mixed Reality (MR)-based systems were proposed. Fonnet (2017) visualized the information stored in the HBIM model and database and proposed an MR application that stores the damage information observed by the building manager [24].

However, these applications only augmented basic information of the building or investigation from the HBIM database. Recently AR applications for support of on-site risk management with HBIM system are developed to solve present problems. However, they cannot retrieve and visualize the AR content according to context because their AR metadata cannot reflect user and heritage context. In order to solve the above problems, we propose metadata for context-aware services in VR/AR.

## 3. Methodology

### 3.1. 5W1H Model-Based Metadata for Context-Aware VR/AR Services

The 5W1H model-based metadata has the extensibility to be used in tourism, education, and heritage management applications, especially in heritage sites. It also has interoperability that can be utilized in various software and applications. This research attempts to utilize the proposed metadata for remote diagnosis in VR. The metadata proposed in this study is used for retrieval and visualization of information according to the context in VR that reflects realistic coordinates. Therefore, it can also be used for AR-based on-site diagnosis.

As the VR/AR service is widely used, research on the metadata standard that can increase the reusability and interoperability of VR/AR content is actively being performed. This study proposes a metadata structure to improve the reusability and interoperability of context-aware VR/AR content metadata. The metadata structure consists of POIs, content, and xAnchor elements. It is designed to solve the problem of existing metadata that can only be used in a specific domain and to enhance scalability. We also propose the POI linker and the content linker that represent the relation among POIs and among contents and explain the description of relation in detail. The proposed metadata

structure can contribute to providing a context-aware VR/AR service by defining the elements, property, and relations necessary for context-aware VR/AR service.

The metadata proposed in the previous study has four problems in implementing context-aware VR/AR services. First, there is difficulty in authoring related services and contents because the metadata considering context-awareness is not defined. Second, it can only be used for a specific domain, which has a problem of lack of flexibility. Third, there is no element to define the relation between VR/AR content and it is difficult to provide related content. Finally, sensor information required for a context-aware VR/AR service is not defined in detail. This study proposes the metadata structure to overcome the above problems and proves the metadata structure through a prototype.

We define the content to be used in the augmented reality service as VR/AR scenes, which consists of POI, xAnchor, Content, linker, and linked content. The POI element consists of who (creator), where (location), what (ID, location), and how (volume, accessibility) to comprise the types of properties that can identify the VR/AR content's location and status. The content element consists of who (author and uploader), when (created time and modified time), what (URI-Uniform Resource Identifier, metadata type, and MIME type-internet media type), and how (accessibility and commercial) to find and generate the content that the VR content creator wants (Figure 1).

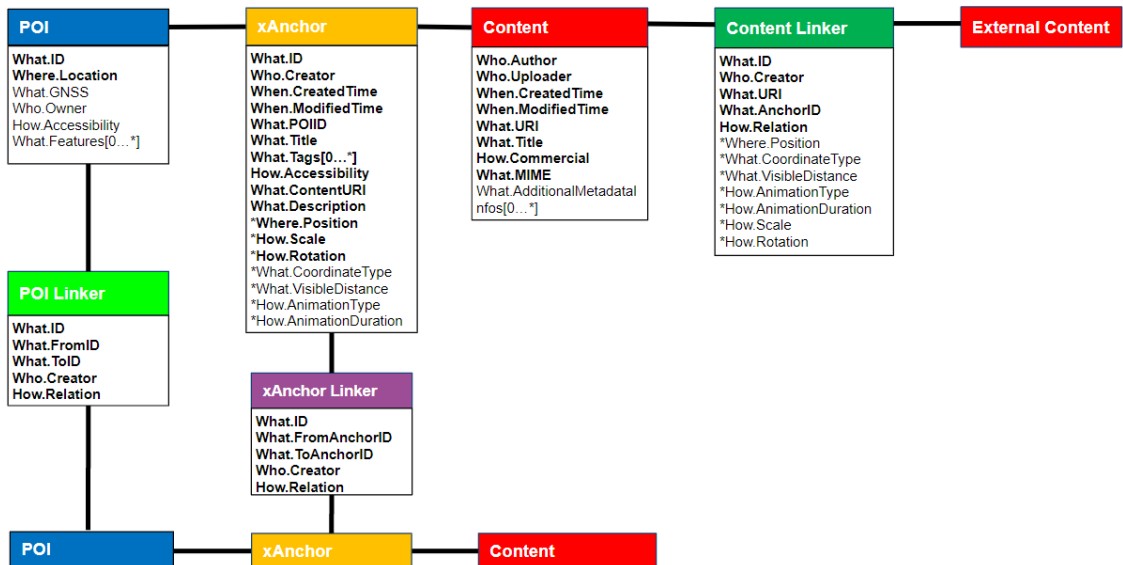

**Figure 1.** Point of Interest (POI)-extended anchor (xAnchor)-content structure based on the 5W1H model-based metadata (The attributes used in this study are shown in bold letters).

The xAnchor element consists of who (creator), when (created time and modified time), where (location), what (ID, POIID, content URI, visible distance, title, description, and coordinate system), and how (animation type, animation period, scale, and rotation) and manages to render methods. Commercial companies, such as Microsoft and Google, defined the anchor as an element that contains only information about location and direction. Microsoft defined the anchor as a common reference frame that allows multiple users to place digital contents in the same physical location [25]. Google explains that an anchor keeps an object in the same location and orientation in space and can maintain the illusion of a virtual object placed in the real world [26]. On the other hand, our proposed xAnchor contains extended information for interactions and content filtering, such as tags.

The proposed metadata contain media representation information in VR. This allows the author of the content to define the content to be experienced by the user specifically. In addition, this study includes a tag for searching POI-xAnchor-content so that users can search for desired information by filtering. In addition, the accessibility of content is added to distinguish it as public, friend, and private, as well as to distinguish commercial versus noncommercial content. The xAnchor elements are useful for expressing, storing, and managing content filtering.

The metadata structure proposed in this study has extensibility to be used in tourism, education, and risk management applications, especially in heritage sites. It also has interoperability that can be utilized in various software and applications. This study attempts to utilize 5W1H model-based metadata for retrieval and visualization of information according to context. The proposed metadata structure contributes to risk management of architectural heritage by being used in VR/AR.

### 3.2. Linker Metadata for Reflecting Domain Knowledge and Retrieving Related Information

We propose the domain ontology that expresses relations among information and relations among POI. We reflect them in the linker metadata. For these metadata, we conceptualized a domain ontology that represents domain knowledge of Korean architectural heritage. This ontology expresses the composition of building components and the cause–effect relationship among components and environmental factors (Figure 2).

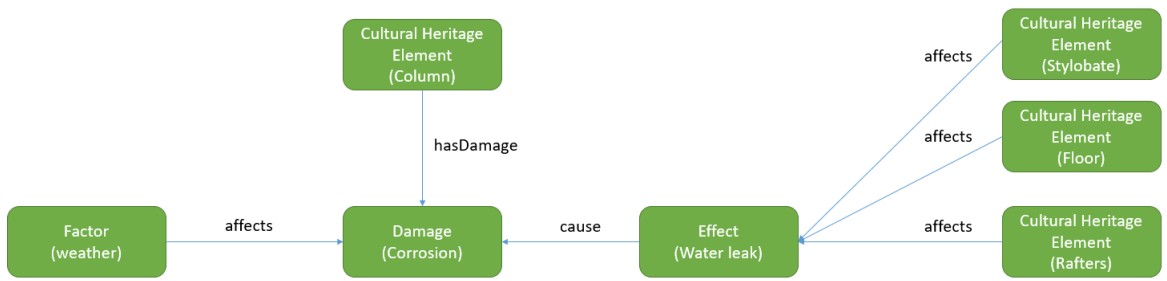

**Figure 2.** Risk ontology for the representation of cause–effect relations among factors and components.

We divided the linker, which plays a role in linking the elements constituting VR/AR, into POI linker, xAnchor linker, and content linker. The POI linker is used to define relationships among real-world coordinates. The xAnchor linker represents the relationship between VR/AR scenes. The xAnchor linker establishes a connection among VR/AR scenes. The content linker is responsible for connecting external content. The content linker metadata, which reflects the risk ontology, defines the contents required for the damage type. The application retrieves information from the external database based on this metadata.

We want to support on-site risk management by linking VR/AR scenes using the xAnchor linker metadata. We designed the ontology for the cause–effect relation of risk and constructed an inference module that applies ontology-based rules. When a client enters a domain ontology-based rule, the visualization of VR/AR scenes' relations can be performed. The xAnchor linker consists of what (ID, FromID, ToID), who (creator), and how (relation) and represent the relation among VR/AR scenes, for example, in the case of column corrosion, connect rafters, foundation, and floor (Figure 3).

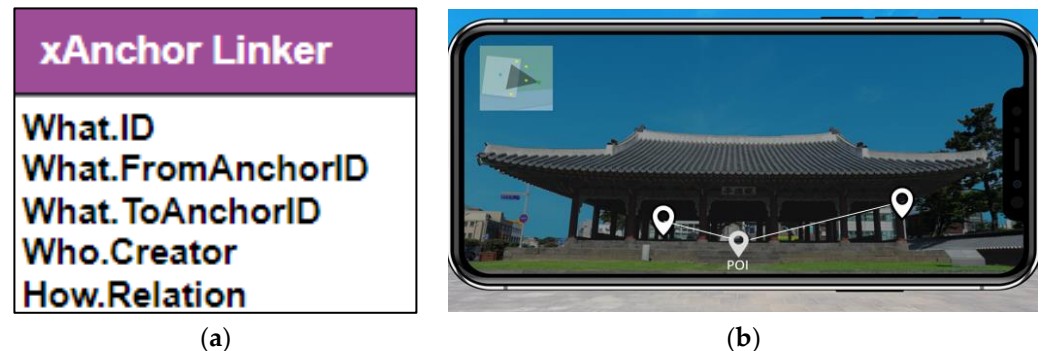

(**a**)                                   (**b**)

**Figure 3.** The xAnchor linker metadata: (**a**) attributes of the xAnchor linker and (**b**) an example of implementation using the xAnchor linker.

We added the content linker metadata that retrieves related information by using domain knowledge-based ontology. Specifically, we defined the content linker metadata as what (ID, FromURL, ToURL, xAnchorID), who (Creator), and how (Relation). We implemented rules to provide and update the data needed to analyze specific damage based on domain ontology. For example, if the column component has corrosion damage, the weather information of the meteorological agency can be retrieved as related information and visualized (Figure 4).

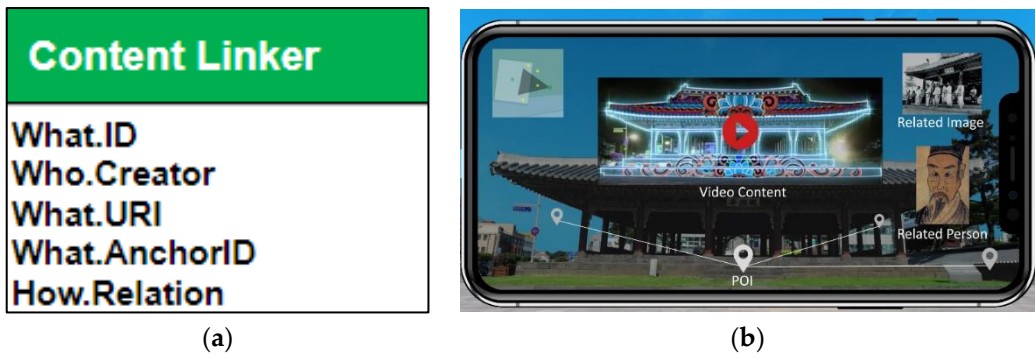

(**a**)　　　　　　　　　　　　　　　　　　　　　　　(**b**)

**Figure 4.** The content linker metadata: (**a**) the content linker metadata and (**b**) an example of implementation using the content linker metadata.

### 3.3. Heritage Building Information Metadata for Risk Management

The heritage building information metadata is divided into building information, component information, and management information classes. The heritage building information package is defined by the 5W1H model showing basic descriptions and properties for risk management. For the definition of the heritage building information content, the building and component class of Korean wooden architectural heritage are defined to represent the structure of architectural heritage and relationships among components and risks (Figure 5).

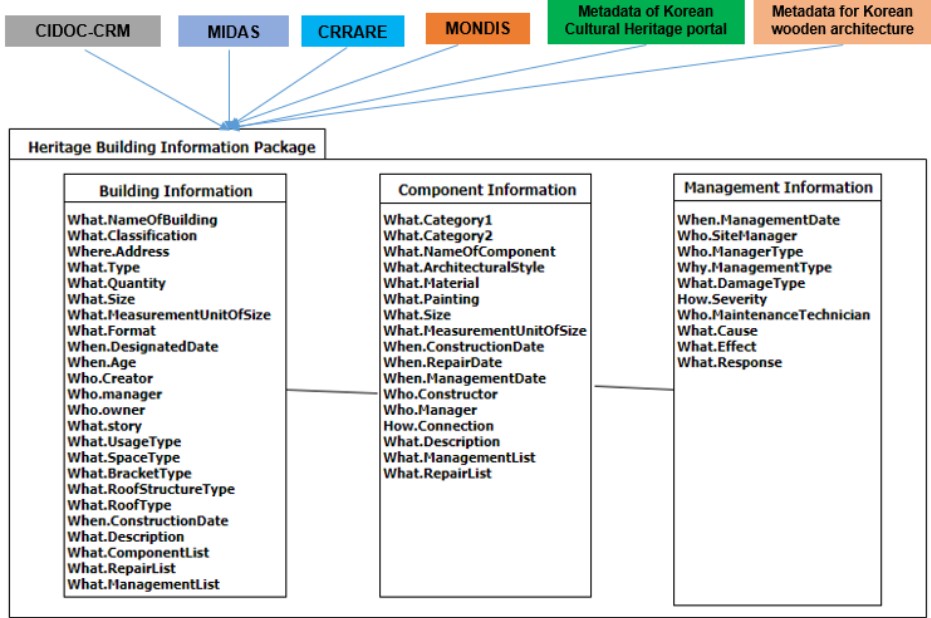

**Figure 5.** Heritage building information metadata.

To define the component taxonomy of Korean wooden architectural heritage which is devoted to description and semantic structuring, we referred to architectural books, repair reports, and expert interviews and we applied the CIDOC Conceptual Reference Model (CRM) standards and vocabularies

which was designed for cultural heritage information and the metadata of a portal system of the Korean Cultural Heritage Administration. For the ontology model, we applied the Monument Damage Information System (MONDIS) to represent the cause–effect relation of damages.

We define building and component class properties of Korean wooden architectural heritage for 4-D description and semantic structuring of architectural heritage. We refer to the 4-D description expression of CIDOC-CRM and the metadata of a portal system of the Korean Cultural Heritage Administration. We referred to the Monument Damage Information System (MONDIS) to represent risk and relation among risks [6].

This metadata describes the detailed risk management information rather than the existing heritage building information metadata. It is advantageous for risk management to check past risk management information and decision making of risk responses. For example, the management information class includes the cause of the damage so that heritage manager can refer to an existing case in the future. This study suggests specialized metadata for risk management of architectural heritage.

Moreover, the heritage building information metadata proposed in this study represent relations between components or information. In particular, Category1 and Category2 are distinguished to express the hierarchy of wooden architectural heritage in East Asia. Category2 means the component group and Category1 means the component type. For example, different components of category 2 often has a cause-effect relation. Also, in the case of management information with the same component type, it is a notable example to understand the cause of damage. The metadata proposed in this study provides heritage information that can be used for risk management in conservation/restoration and relation rather than existing metadata.

### 3.4. Framework of Remote Diagnosis in Virtual Reality

The site manager and nonexpert use the mobile Geographic Information System (GIS)-based geo-tagged content tool in the field to upload pictures of a heritage building's damage, descriptions, and tags. Specifically, pictures contain tags that include the uploader's expertise, kind of damage, and severity that are attributes of the heritage building information metadata. This information is stored in a geo-tagged content database by a geospatial server. The geo-tagged database stores and manages on-site information that is accumulated, such as images and texts.

The expert on the remote site uses the Head Mounted Display (HMD) and connects to the remote VR application to enter the domain expertise and tasks. The server receives the user information (domain expertise and task) and retrieves content from the geo-tagged content database that contains the tag appropriate for the user information. Content includes historical and on-site information that is stored in the HBIM database.

The filtered content is provided to the expert according to selected expertise and task. If the expert selects his or her task and domain expertise, the application provides the risk management information according to the selected task and domain expertise. Experts can see all of the information needed for monitoring in a virtual environment, but too much information can overload to experts.

Among the filtered contents, the POI including the management information is arranged by color (green, yellow, and red) according to severity. The expert can determine the order of components by visualizing the condition color and the connection relationship of the component that is related to the damage of the component. The expert can also check related information with the linker content metadata according to the tags of uploaded information. Finally, the expert checks the diagnosis results in the checklist and stores them in the geo-tagged content database (Figure 6).

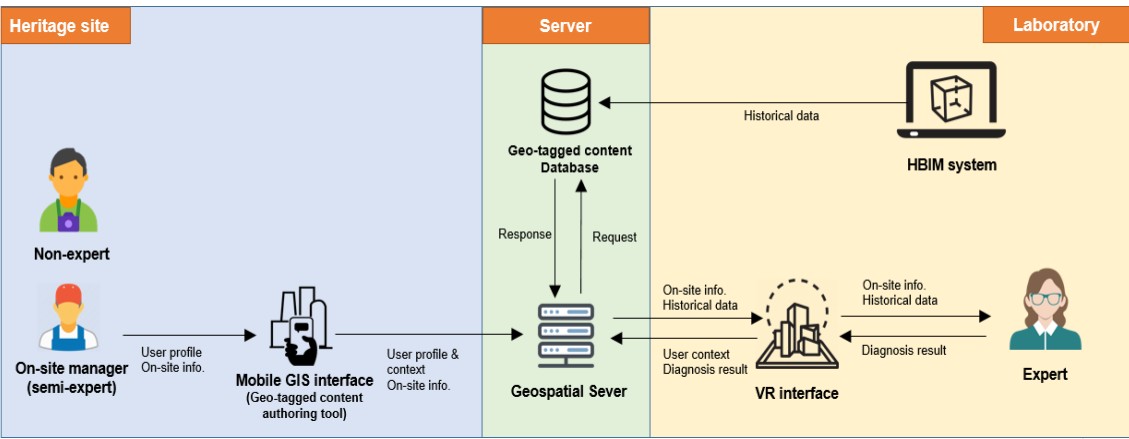

**Figure 6.** A framework of remote diagnosis in virtual reality.

## 4. Implementation

The remote diagnosis framework that is proposed in this study is divided into a geo-tagged content authoring tool and a remote diagnosis application. First, the geo-tagged authoring tool allows the site manager to upload daily management information (checklists, site photos, and comments) regarding actual heritage. Second, a remote diagnosis VR application enables experts to diagnose virtual heritage. The geo-tagged content authoring is created based on Cesium for the 3-D mapping of geospatial data. Cesium is an open-source JavaScript library for world-class 3-D mapping of geospatial data. The application can be worked in on a desktop or mobile environment. The site manager can generate POI; uploads site photos; and enters a title, description, and tags into the 5W1H model-based metadata using this application (Table 1).

**Table 1.** Example of POI-xAnchor-content metadata in a geo-tagged content authoring tool.

| Category | 5W1H | Metadata Name | Example |
|---|---|---|---|
| **POI** | **What** | **ID** | 22 |
| | **Where** | **Location** | "latitude": 33.51330, "longitude":126.52152, "altitude": 39.0 |
| **xAnchor** | **What** | **ID** | 147 |
| | **Who** | **Creator** | Jack |
| | **When** | **Created Time** | 1 April 2019 |
| | **When** | **Modified Time** | 2 April 2019 |
| | **What** | **POI ID** | 22 |
| | **Where** | **Content Location** | "positionx": 1, "positiony": 1, "positionz": 1 "scalex": 1, "scaley": 1, "scalez": 1 "locationx": 1, "locationy": 1, "locationz": 1 |
| | **What** | **Title** | Column2 management information |
| | **What** | **Tag** | 1 April 2019, Column (what), Serious (how), Corrosion (what), Monitoring (why) |
| | **What** | **Description** | Maintenance technician: Brian, Response: Repair & maintenance |
| | **What** | **Accessibility** | Public |
| | **What** | **Content URI** | https://s3.ap-northeast-2.amazoneaws.com/kctm/server3/1/column.jpg |
| **Content** | **Who** | **Author** | John |
| | **Who** | **Uploader** | John |
| | **When** | **Created Time** | 2 April 2019 |
| | **When** | **Modified Time** | 2 April 2019 |
| | **What** | **URL** | https://s3.ap-northeast-2.amazoneaws.com/kctm/server3/1/column.jpg |
| | **What** | **Title** | Column photo |
| | **How** | **Commercial** | Noncommercial |
| | **What** | **Internet media types (MIME types)** | image/jpeg |

The site manager uses a web-based geo-tagged content authoring tool that supports geo-tagging, the uploading of content, site photos and comments, and the entry of content metadata and tags. The site manager creates POIs and geo-tags on a three-dimensional map. The user can also take pictures at the site, upload them, and conduct geo-tagging of the status of the component (good, normal, and serious) and the manager's name. The description is a comment, which means information on the content. The user can also enter accessibility (public, friend, and private) (Figure 7).

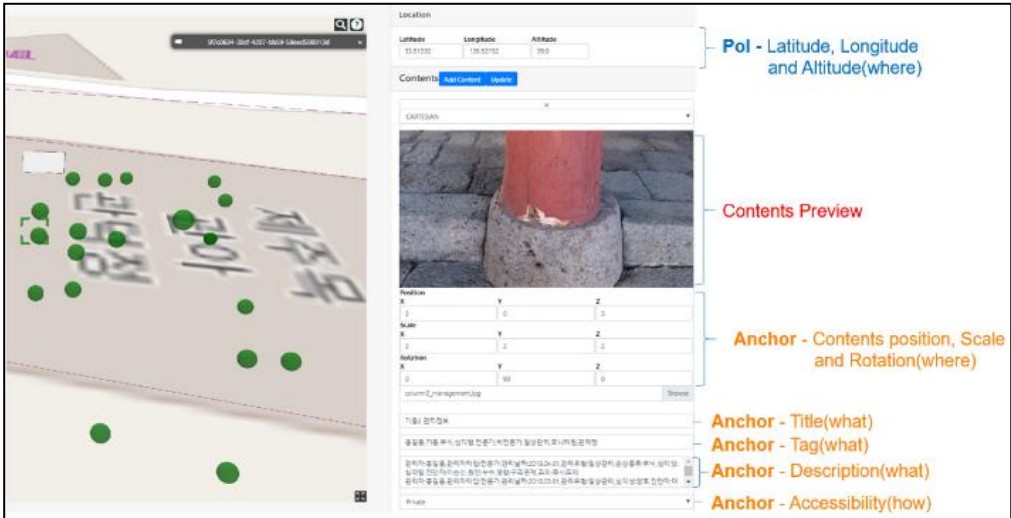

**Figure 7.** A geo-tagged content authoring tool.

We used a Computer Aided Design (CAD)-based modeling and photogrammetry to construct a virtual model of Gwandeokjeong building on Jeju Island. Photogrammetry is the technology of determining the size, shape, and identification of objects by analyzing terrestrial or aerial imagery. The photogrammetric processing includes several stages which allow the generation of 3-D digital models. We took 20 pictures of column 2 and created the photogrammetry 3-D model using ReCap software (Figure 8). Then, we manually deleted unnecessary meshes of the 3-D model acquired using photogrammetry using 3ds Max. Then, we converted them into FBX file format. The front column of the building reflects the realistic texture, especially column 2, which is made with photogrammetry and reflects its real appearance for the evaluation of column We added the virtual building to a VR testbed based on the Unity3D game engine for the remote diagnosis application.

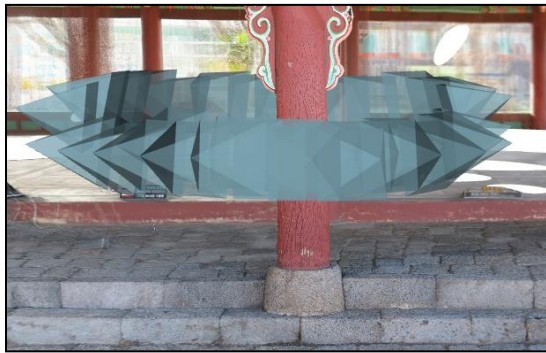

**Figure 8.** A photogrammetry 3-D model of column2.

The remote diagnosis VR application is developed by the Unity 3D game engine. The Unity 3D game engine provides a user interface to create a VR application and to connect the geo-tagged spatial database by the server. The VR-based remote diagnosis application was built on Android to run on the HMD VR device. This application contains the functions specified by the remote expert.

The application filters and visualizes the on-site management information according to the user's domain expertise and task (Figure 9).

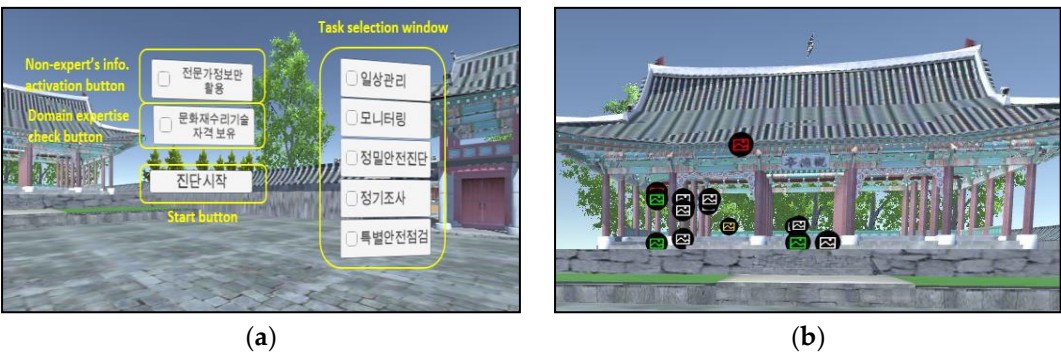

**Figure 9.** An example of contents filtering according to domain expertise and task: (**a**) a log-in screenshot and (**b**) filtered contents of remote diagnosis application.

It also provides information on the external database via content linker metadata and presents POI relations through the xAnchor linker metadata. For example, it retrieves and visualizes meteorological analysis information from the meteorological database (Figure 10). The expert analyzes the risk by referring to the filtered contents and determines the response.

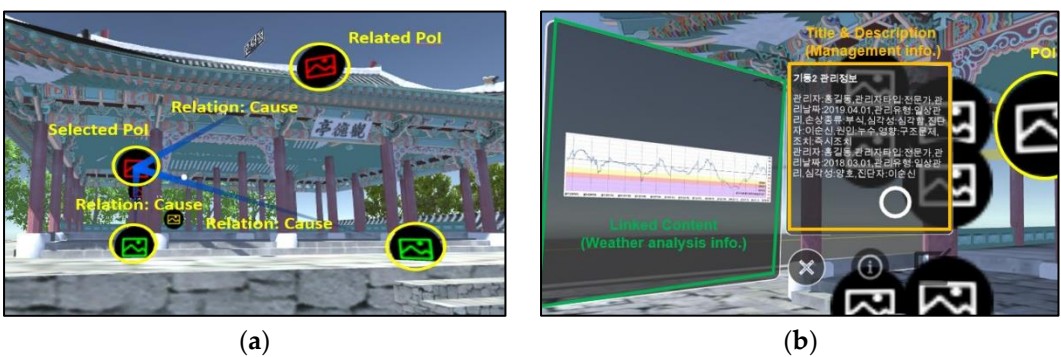

**Figure 10.** An example of using linker metadata: (**a**) an example of implementation of the xAnchor linker in remote diagnosis in virtual reality (VR) and (**b**) an example of implementation of the content linker in remote diagnosis in VR.

## 5. Evaluation

### 5.1. Experiment Design for Remote Diagnosis in Virtual Reality

We conducted a focus-group interview and a heuristic evaluation with experts to explore the usefulness of remote diagnosis in VR. First, the experts discussed the current status and limitations of existing remote diagnosis methods based on photographs and reports. Second, we evaluated advantages and disadvantages of the remote VR diagnosis in the focus-group interview. Finally, we conducted a heuristic evaluation of the context, performance, interaction, and usability of the proposed application as an interactive information retrieval system.

The experiment to verify the proposed application was conducted on April 18, 2019. Four members of the monitoring team of the Daejeon cultural heritage management agency with cultural heritage repair qualifications participated. For remote diagnosis in VR, Lenovo Mirage Solo with the Google Daydream platform was used (Figure 11). The experiment consisted of 20 minutes of experiment explanation, 20 minutes of focus-group interview, and 60 min of heuristic evaluation and group interview. Two tasks were performed to diagnose daily management information and to monitor

information of column 2 on Jeju Island using the application. All the content during the focus-group interview was recorded and documented.

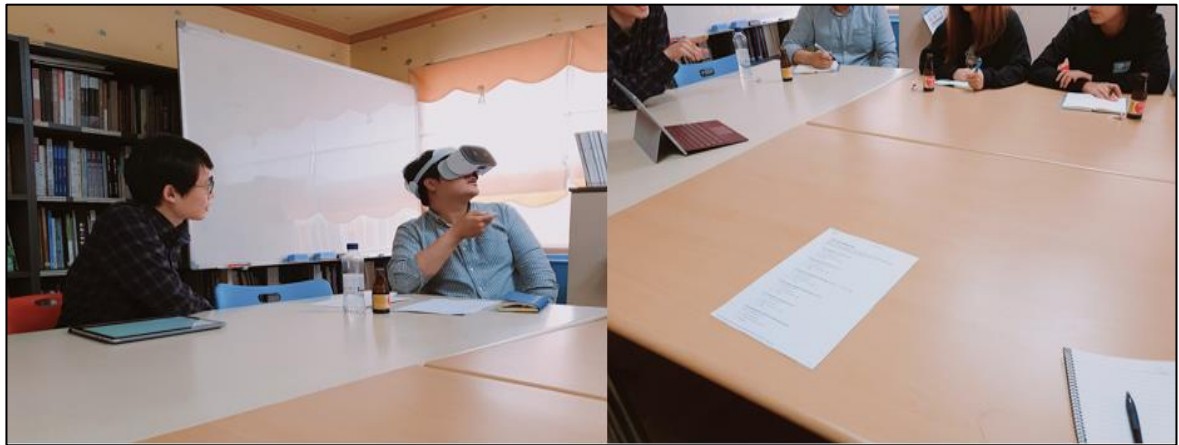

**Figure 11.** Photos of heuristic evaluations.

A heuristic evaluation is a usability inspection method that uses a short list of heuristic guidelines and only a few evaluators in the Human–Computer Interaction (HCI) domain [27]. This evaluation method has the advantage of obtaining quick and easy feedback to improve a system. Each evaluator is given a short list of heuristic principles; then, he or she uses the system to identify a usability problem. After the individual test, the evaluators communicated with each other about the system usability. In this study, we designed 11 heuristics, as shown in Table 2. We refer to the evaluation criteria of the interactive information retrieval system and reflect upon it when creating the heuristic principles [28].

**Table 2.** Heuristic evaluation criteria.

| No. | Criteria | | Description |
| --- | --- | --- | --- |
| 1 | Context | | Provision of information according to purpose and task |
| | | | Provision of information according to behavior |
| 2 | Performance | | Operational error |
| | | | System error |
| 3 | Interaction | | The method and process to confirm information for the task |
| 4 | Usability | Ease of learning | The degree to understand task |
| 5 | | Ease of use | The degree to distinguish or identify information |
| 6 | | Effectiveness | Accuracy and completeness with which users achieve specified goals |
| 7 | | Efficiency | Resources expended in relation to the accuracy and completeness with which users achieve goals |
| 8 | | Flow | Mental state of operation in which a person is fully immersed in what he she is doing, characterized by a feeling of energized focus, full involvement, and success in the process of the activity |
| 9 | | Engagement | A quality of user experiences with technology that is characterized by challenge, aesthetic and sensory appeal, feedback, novelty, interactivity, perceived control and time, awareness, motivation, and interest and affect |
| 10 | | Satisfaction | Freedom from discomfort and positive attitudes of the user to the product |
| 11 | | Preference | Evaluation of providing information and interfaces based on preference |

The principles can be divided by context, performance, interaction, and usability. Usability consists of ease of learning, ease of use, effectiveness, efficiency, flow, engagement, satisfaction, and preference. The context means whether the information is provided according to age, sex, task type,

or domain expertise. The performance is related to operational and system errors. The interaction refers to the evaluation of the activities and the processes of an interactive information retrieval system. Finally, the usability means a quality attribute that assesses user interfaces [28].

*5.2. Result of Experiment*

5.2.1. Focus-Group Interview on Existing Remote Diagnosis

Experts explain the process of deciding the existing remediation of cultural heritage and future measures and discuss limitations. The existing remote diagnosis of cultural heritage is carried out in the office by managers (site managers, maintenance technicians, and office workers). They referred to photographs, related materials, historical data, and repair reports. Then, they confirmed photos, drawings, and checklists to analyze risk and to make decisions regarding risk response.

One expert commented that past risk management information such as maintenance history is not currently used but that it should be provided as a reference for remote diagnosis. The experts discussed the usefulness of the proposed VR-based diagnosis compared to the existing remote diagnosis methods. An expert said it would be helpful to be able to see information from a variety of sources and formats at once. Another expert said that the proposed application is expected to be able to identify fast work and to verify the connection through the Internet.

In the focus-group interview regarding remote diagnosis in VR, the experts proposed four improvements. First, on-site information should be updated immediately. Second, experts felt that it is important for the specialists performing the diagnosis to be provided with only the necessary information. Third, it is important to show the sequence of information to be checked. Fourth, the experts believed that providing environmental information and data is important. In this study, we implemented an open GIS-based on-site management system, content filtering, system visualization, and metadata for providing external data in order to provide environmental information and data from an external database.

- It is important to update the on-site information frequently.
- Basic information is not necessary because experts have domain expertise.
- It is essential to set priorities of which components should be checked in the application.
- Environmental data (humidity, temperature, etc.) of the heritage site should be recorded and provided.

They said that they could use the manager's on-site information as an open GIS-based system in real time so that they could take immediate action. In addition to providing filtered content according to task and domain expertise, it also provided external information, such as weather analysis details, to help us to determine the cause of the damage. They also said that it is easy to determine the diagnostic sequence by visualizing information according to the severity of the condition. Finally, the provision of environmental information, such as weather information, will help to determine the diagnosis and action. Overall, all of the experts on the system improvements responded positively and said that such changes would contribute to reducing the time required for remote diagnosis and improve accuracy. According to Table 3, they proposed suggestions to improve the application as well as the advantages and disadvantages of the application.

**Table 3.** Results of the focus-group interviews.

| Expert No. | Age/Sex | Career (Year) | Advantage | Disadvantage | Suggestion |
|---|---|---|---|---|---|
| 1 | 20–30/Female | 3 | Showing the priority of diagnosis | Difficulty of navigation and checking images | A more intuitive interface for navigation and confirmation of images |
| 2 | 30–40/Female | Over 10 | Providing external information | Difficulty of navigation and checking images | A record method for the positional movement of components |
| 3 | 30–40/Male | Over 10 | Supporting to understand the situation faster | Difficulty of navigation and checking images | A hybrid method with existing diagnosis methods (photos, drawings, and reports) |
| 4 | 30–40/Female | Over 10 | Viewing information from a variety of sources and formats | Dizziness | A function to check photos in chronological order. |

### 5.2.2. Heuristic Evaluations on Remote Diagnosis in VR

Four maintenance technicians participated in the heuristic evaluations of the application. Three of the four experts have over ten years of cultural heritage management experience, and one has three years of experience. They are all certified by the government for cultural heritage repair. Familiarity with the virtual reality environment is more prevalent than usual except for one expert. The experts spent an average of 30 minutes of assessment time. After reviewing the system as a whole, experts answered a scale of 1 (no problem) to 5 (major problem) for the given issue for this experiment (Figure 12). Additionally, they discussed each issue through a focus-group interview after performing a heuristic evaluation. Issues are context, performance, interaction, ease of learning, ease of use, effectiveness, efficiency, flow, engagement, satisfaction, and preference.

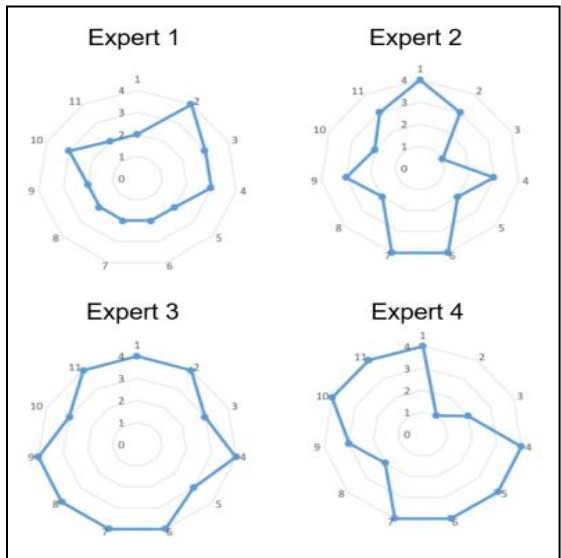

**Figure 12.** Results of heuristic evaluations.

In an interview on the context topic, all experts were satisfied with providing information according to task and domain expertise. However, one expert gave relatively negative feedback about information according to domain knowledge. We can think that the proposed application classifies domain expertise into two steps. These results show that domain expertise needs more various levels. Also, expert No. 1 said that it was necessary to improve the information provided by behavior. However, the information is not provided according to the behavior in the virtual environment. Therefore, it is necessary to improve it as a context-aware service that recognizes the user's behavior in the future.

Expert No. 4 commented that there was an error in image loading. This is probably due to a problem in downloading information from the geospatial database due to network problems. Expert No. 2 experienced a failure to load a photo due to the system and network errors. Expert No. 1 replied that he experienced receiving unintended information. We plan to verify these figures quantitatively in the future because there was a lack of the number of subjects and contents.

In the information retrieval system, interaction means how many processes an expert performs to complete a task. The experts interacted with content on an average of six times to perform tasks. Expert No. 2 was dissatisfied with a lot of interactions to complete the task. In order to obtain accurate diagnosis results while reducing the number of interactions, it is necessary to filter the necessary information according to the task.

Discussions on usability are as follows. There was an opinion that all experts would like to add a more intuitive interface for movement and information confirmation. For example, expert No. 2 said that there was an opinion that the information should be easily identifiable when confirming photographs or graphs. There was also a demand for zoom in/zoom out capabilities and a way to enlarge photographs. Expert No. 3 said that checking the status in chronological order is very important.

The experts discussed the following suggestions. Expert No. 4 mentioned that the task should provide more detail content. For example, there was an opinion that detailed information such as repair details is not utilized among the information filtered by the daily management. There was also a need to add a method to measure and record the displacements of components easily. Finally, expert No. 3 commented that remote diagnosis in VR would be more helpful when the existing methods are used together.

## 6. Conclusions

We proposed a VR-based remote diagnosis which enables the retrieval and visualization of geospatial information according to the context to support maintenance technicians' diagnosis at the remote site. They can analyze on-site information and decision making for a response in VR. This study is expected to reduce the time and expense of the field diagnosis through the remote diagnosis in VR, to reduce the diagnosis time, and to increase accuracy compared to the existing remote diagnosis.

The focus group interview shows that maintenance technicians are satisfied with the diagnostic sequence information, external information for reference, and overall state information of the building. This interview also shows that they have difficulty in navigating and viewing the information in the VR and feel dizziness. Therefore, we can plan to improve an interface and a navigation method.

The result of heuristic evaluation shows that maintenance technicians were most satisfied with the provision of contextualized information (avg. 3.5), ease of learning, effectiveness, and efficiency of the application. Their high score on the context aspect seems to be due to the remote diagnosis application that filters and visualizes the information they need. In addition, we think that this function is effective and efficient because it enables help and support of diagnosis work. The application of this study intuitively provides historical and related information for remote diagnosis, which is considered highly satisfactory for ease of use. On the other hand, they find it difficult to navigate in a virtual environment and seem to be most unsatisfied with performance (avg. 2.25).

The remaining work of this study is arranged as follows. Site managers have to upload photos and enter tags manually. We will make it possible to enable automatic tagging of uploaded photos and contents in the future. Experts will also be able to filter and retrieve tagged information as well as to verify data through the POI choice. As experts suggested in the focus-group interview, we will add a more intuitive way to check on-site information and an interface to identify past field information.

The future work of this study is as follows. First, we intend to develop the AR-based on-site risk management tool that can provide object recognition, information retrieval, and provision through tracking. We will compare the performance of tasks with those of experts and nonexperts. We will also interview them to see what aspects they consider important in using this tool. Second, we will develop a crowdsourcing platform that can induce public participation by adding appropriate compensation in

the future. Third, we want to develop a context-aware system that can reduce information overload by providing information by analyzing user behavior. Fourth, we will build a digital twin that synchronizes sensing information and visual information in real time and enables predictive maintenance.

**Author Contributions:** Conceptualization, J.L. and W.W.; Funding acquisition, W.W.; Investigation, J.L.; Methodology, J.L., J.K. and W.W.; Project administration, W.W.; Supervision, W.W.; Visualization, J.K.; Writing—original draft, J.L. and J.K.; Writing—review & editing, J.A. and W.W.

**Funding:** This research received no external funding.

**Acknowledgments:** This research is supported by Ministry of Culture, Sports, and Tourism (MCST) and Korea Creative Content Agency (KOCCA) in the Culture Technology (CT) Research & Development Program 2017.

**Conflicts of Interest:** The authors declare no conflict of interest.

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
