# Peer review of "Remote Diagnosis of Architectural Heritage Based on 5W1H Model-Based Metadata in Virtual Reality"

_ijgi, doi:10.3390/ijgi8080339_

Round 1

Reviewer 1 Report

The research regards the application of a tool for diagnostic monitoring of cultural heritage with remote control and based on augmented reality.

The structure of the text is fine, as is the methodological development and implementation. The case study description is a bit reductive both from content point of view and level of complexity presented.

Some aspects of the research, on the other hand, are unclear.

First of all, the acronym of the 5W1H project is based on who, what, why, when, where, how, but in all the schemes there is not “why”. So is better to clarify this aspect or to modify the acronym.

The use of POIs seems quite intuitive, but at the process level it is only tested by expert or semi-expert users, while it would be interesting to verify its use by non-expert users. If this evaluation has not been planned in the heuristic evaluation phase, it would be good to plan it in the future, as it would be good to understand what different contribution is expected from the two different figures (non-experts and semi-experts) and if the two channels are filtered differently by the experts in the laboratory phase. Moreover, the planning of an evaluation test to a wider audience of people would make the evaluation statistically more significant.

It is also suggested to better specify the construction phase of the model and how it is translated into a mathematical model. In fact, we move from a brief description of the image-based modeling of a column to the display in unity of a complex model. How are the models linked together? Are they both the result of a common parametrization operation? The virtual model, which represents the end-user interface, is very important in the interaction phase, so how it was built and what is its level of reliability in metric terms compared to the real example?

The conclusions are a bit synthetic, it would be good to resume the experimentation and draw conclusions from the collected data, as a useful moment of synthesis of the experimentation.

Finally, some brief notes on some errors or forgetfulness at the grammatical level:

pg 1, line 37 - repetition of the word "management" in a nutshell, finding a synonym;

pg 2, line 199 - "location, status" probably some words are missing in the sentence;

pg 7, line 273 - "We ... the ontology model" is missing the verb of the sentence;

pg 9, line 327 - "PoI" is used from here on, alternating with "POI". Standardise the acronym in writing;

pg 16, line 473 - "the task In order to" misses the point after "task".

Before publishing the article, it would be very important to clarify the progress of this publication with respect to the content published by the same authors in the reference [14], since much information, the structure and the case study appear to be the same (https://www.sciencedirect.com/science/article/pii/S1296207418304692). If no real progress is demonstrated, it makes no sense for the article to be published. For this reason in this moment a major revision has been required, waiting for a specification by the authors about this stuff.

Reviewer 2 Report

Dear authors,

The authors proposed a VR-based remote diagnosis which enables to retrieve and visualize geospatial information according to the context to support an expert’s diagnosis at the remote site. They conducted a focus group interview to evaluate the proposal.

In general, the paper needs to be checked by an English speaker. Some parts need to be rewritten using a more scientific style. Some notes are added in the attached pdf. These corrections have to be solved before the paper publication

Best regards

Reviewer 3 Report

The paper is well written, clear and interesting.

Only a few suggestions:

line 21 to 24: the sentence is not very clear.

line 46: "across through various" maybe only "through"

line 55:  "is useful a technology" should become "is a useful  technology"

 line 153 to 155: the sentence is not clear

line 174:  "contezt" become "context" 

line 197 and line 202: is written Who (creator) and in the image is written who (owner) . It is better to unify the definition

line 220: "reflecting of ..." maybe better "reflecting domain"...

line 259 and line 263 is not fig.24 but fig 5

line 273:verb is missing in the sentence

line 309: the sentence is not clear

line 473 the verb should be "was" singular

line 494 it is not clear...do you want to reduce the accuracy? it sounds strange!

Round 2

Reviewer 1 Report

As a consequence of author’s corrections, the article can be published with minor English language check.

Author Response

I completed the check and rephrase sentences according to the review.

Reviewer 2 Report

The authors manages all reviewer comments. The paper is ready to publish.

Author Response

I completed the check and rephrase sentences.
